# Efficient summarization with read-again and copy mechanism

**Wenyuan Zeng[†], Wenjie Luo[‡], Sanja Fidler[‡], Raquel Urtasun[‡]**
[†]Tsinghua University, [‡]University of Toronto
`cengwy13@mails.tsinghua.edu.cn`
`{wenjie, fidler, urtasun}@cs.toronto.edu`

## Abstract

Encoder-decoder models have been widely used to solve sequence to sequence prediction tasks. However current approaches suffer from two shortcomings. First, the encoders compute a representation of each word taking into account only the history of the words it has read so far, yielding suboptimal representations. Second, current models utilize large vocabularies in order to minimize the problem of unknown words, resulting in slow decoding times and large storage costs. In this paper we address both shortcomings. Towards this goal, we first introduce a simple mechanism that first reads the input sequence before committing to a representation of each word. Furthermore, we propose a simple copy mechanism that is able to exploit very small vocabularies and handle out-of-vocabulary words. We demonstrate the effectiveness of our approach on the Gigaword dataset and DUC competition outperforming the state-of-the-art.

## 1 Introduction

Encoder-decoder models have been widely used in sequence to sequence tasks such as machine translation (Cho et al. (2014); Sutskever et al. (2014)). They consist of an encoder which represents the whole input sequence with a single feature vector. The decoder then takes this representation and generates the desired output sequence. The most successful models are LSTM and GRU as they are much easier to train than vanilla RNNs.

In this paper we are interested in summarization where the input sequence is a sentence/paragraph and the output is a summary of the text. Several encoding-decoding approaches have been proposed (Rush et al. (2015); Hu et al. (2015); Chopra et al. (2016)). Despite their success, it is commonly believed that the intermediate feature vectors are limited as they are created by only looking at previous words. This is particularly detrimental when dealing with large input sequences. Bi-directorial RNNs (Schuster & Paliwal (1997); Bahdanau et al. (2014)) try to address this problem by computing two different representations resulting of reading the input sequence left-to-right and right-to-left. The final vectors are computed by concatenating the two representations. However, the word representations are computed with limited scope.

The decoder employed in all these methods outputs at each time step a distribution over a fixed vocabulary. In practice, this introduces problems with rare words (e.g., proper nouns) which are out of vocabulary. To alleviate this problem, one could potentially increase the size of the decoder vocabulary, but decoding becomes computationally much harder, as one has to compute the soft-max over all possible words. Gulcehre et al. (2016), Nallapati et al. (2016) and Gu et al. (2016) proposed to use a copy mechanism that dynamically copy the words from the input sequence while decoding. However, they lack the ability to extract proper embeddings of out-of-vocabulary words from the input context. Bahdanau et al. (2014) proposed to use an attention mechanism to emphasize specific parts of the input sentence when generating each word. However the encoder problem still remains in this approach.

In this work, we propose two simple mechanisms to deal with both encoder and decoder problems. We borrowed intuition from human readers which read the text multiple times before generating summaries. We thus propose a 'Read-Again' model that first reads the input sequence before committing to a representation of each word. The first read representation then biases the second read

representation and thus allows the intermediate hidden vectors to capture the meaning appropriate for the input text. We show that this idea can be applied to both LSTM and GRU models. Our second contribution is a copy mechanism which allows us to use much smaller vocabulary sizes resulting in much faster decoding and much smaller storage space. Our copy mechanism also allows us to construct a better representation of out-of-vocabulary words. We demonstrate the effectiveness of our approach in the challenging Gigaword dataset and DUC competition showing state-of-the-art performance.

## 2 RELATED WORK

### 2.1 SUMMARIZATION

In the past few years, there has been a lot of work on extractive summarization, where a summary is created by composing words or sentences from the source text. Notable examples are Neto et al. (2002), Erkan & Radev (2004), Wong et al. (2008), Filippova & Altun (2013) and Colmenares et al. (2015). As a consequence of their extractive nature the summary is restricted to words (sentences) in the source text.

Abstractive summarization, on the contrary, aims at generating consistent summaries based on understanding the input text. Although there has been much less work on abstractive methods, they can in principle produce much richer summaries. Abstractive summarization is standardized by the DUC2003 and DUC2004 competitions (Over et al. (2007)). Some of the prominent approaches on this task includes Banko et al. (2000), Zajic et al. (2004), Cohn & Lapata (2008) and Woodsend et al. (2010). Among them, the TOPIARY system (Zajic et al. (2004)) performs the best in the competitions amongst non neural net based methods.

Very recently, the success of deep neural networks in many natural language processing tasks (Collobert et al. (2011)) has inspired new work in abstractive summarization . Rush et al. (2015) propose a neural attention model with a convolutional encoder to solve this task. Hu et al. (2015) build a large dataset for Chinese text summarization and propose to feed all hidden states from the encoder into the decoder. More recently, Chopra et al. (2016) extended Rush et al. (2015)'s work with an RNN decoder, and Nallapati et al. (2016) proposed an RNN encoder-decoder architecture for summarization. Both techniques are currently the state-of-the-art on the DUC competition. However, the encoders exploited in these methods lack the ability to encode each word condition on the whole text, as an RNN encodes a word into a hidden vector by taking into account only the words up to that time step.

In contrast, in this work we propose a 'Read-Again' encoder-decoder architecture, which enables the encoder to understand each input word after reading the whole sentence. Our encoder first reads the text, and the results from the first read help represent the text in the second pass over the source text. Our second contribution is a simple copy mechanism that allows us to significantly reduce the decoder vocabulary size resulting in much faster inference times. Furthermore our copy mechanism allows us to handle out-of-vocabulary words in a principled manner. Finally our experiments show state-of-the-art performance on the DUC competition.

### 2.2 NEURAL MACHINE TRANSLATION

Our work is also closely related to recent work on neural machine translation, where neural encoder-decoder models have shown promising results (Kalchbrenner & Blunsom (2013); Cho et al. (2014); Sutskever et al. (2014)). Bahdanau et al. (2014) further developed an attention mechanism in the decoder in order to pay attention to a specific part of the input at every generating time-step. Our approach also exploits an attention mechanism during decoding.

### 2.3 OUT-OF-VOCABULARY AND COPY MECHANISM

Dealing with Out-Of-Vocabulary words (OOVs) is an important issue in sequence to sequence approaches as we cannot enumerate all possible words and learn their embeddings since they might not be part of our training set. Luong et al. (2014) address this issue by annotating words on the source, and aligning OOVs in the target with those source words. Recently, Vinyals et al. (2015)

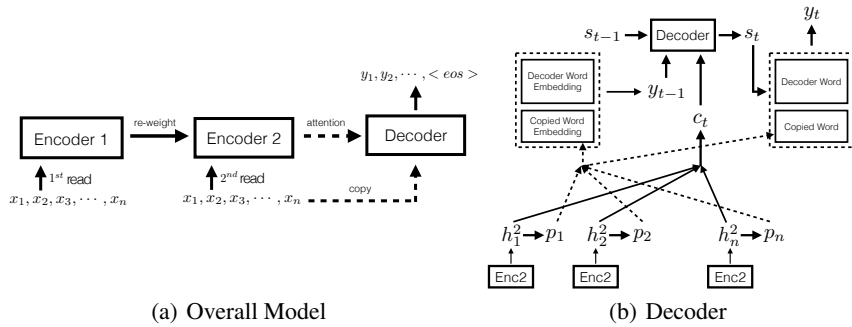

Figure 1: Read-Again Summarization Model

propose Pointer Networks, which calculate a probability distribution over the input sequence instead of predicting a token from a pre-defined dictionary. Cheng & Lapata (2016) develop a neural-based extractive summarization model, which predicts the targets from the input sequences. Gulcehre et al. (2016); Nallapati et al. (2016) use explicit gating to decide adaptively wether to generate a target word from the fixed-size dictionary or from the input sequence. Gu et al. (2016) use a implicit implicit gating operation instead of the explicit gating. This is similar to our decoder. However, our decoder can also extract different OOVs' embedding accordingly from the input text instead of using a single <UNK> embedding to represent all OOVs. This further enhances the model's ability to handle OOVs.

## 3 THE READ AGAIN MODEL

Text summarization can be formulated as a sequence to sequence prediction task, where the input is a longer text and the output is a summary of that text. In this paper we develop an encoder-decoder approach to summarization. The encoder is used to represent the input text with a set of continuous vectors, and the decoder is used to generate a summary word by word.

In the following, we first introduce our 'Read-Again' model for encoding sentences. The idea behind our approach is very intuitive and is inspired by how humans do this task. When we create summaries, we first read the text and then we do a second read where we pay special attention to the words that are relevant to generate the summary. Our 'Read-Again' model implements this idea by reading the input text twice and using the information acquired from the first read to bias the second read. This idea can be seamlessly plugged into LSTM and GRU models. Our second contribution is a copy mechanism used in the decoder. It allows us to reduce the decoder vocabulary size dramatically and can be used to extract a better embedding for OOVs. Fig. 1(a) gives an overview of our model.

### 3.1 ENCODER

We first review the typical encoder used in machine translation (e.g., Sutskever et al. (2014); Bahdanau et al. (2014)). Let $x = \{x_1, x_2, \cdots, x_n\}$ be the input sequence of words. An encoder sequentially reads each word and creates the hidden representation $h_i$ by exploting a recurrent neural network (RNN)

$$h_i = \text{RNN}(\mathbf{x_i}, h_{i-1}), \tag{1}$$

where $\mathbf{x_i}$ is the word embedding of $x_i$. The hidden vectors $h = \{h_1, h_2, \cdots, h_n\}$ are then treated as the feature representations for the whole input sentence and can be used by another RNN to decode and generate a target sentence. Although RNNs have been shown to be useful in modeling sequences, one of the major drawback is that $h_i$ depends only on past information i.e., $\{x_1, \cdots, x_i\}$. However, it is hard (even for humans) to have a proper representation of a word without reading the whole input sentence.

Following this intuition, we propose our 'Read-Again' model where the encoder reads the input sentence twice. In particular, the first read is used to bias the second more attentive read. We apply

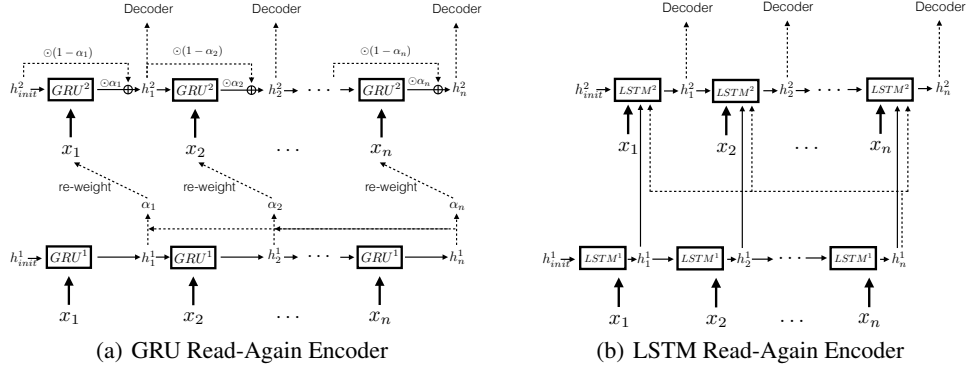

(a) GRU Read-Again Encoder (b) LSTM Read-Again Encoder

Figure 2: Read-Again Model

this idea to two popular RNN architectures, i.e. GRU and LSTM, resulting in better encodings of the input text. Note that although other alternatives, such as bidirectional RNN exist, the hidden states from the forward RNN lack direct interactions with the backward RNN, and thus forward/backward hidden states still cannot utilize the whole sequence. Besides, although we only use our model in a uni-directional manner, it can also be easily adapted to the bidirectional case. We now describe the two variants of our model.

### 3.1.1 GRU READ-AGAIN

We read the input sentence $\{x_1, x_2, \cdots, x_n\}$ for the first-time using a standard GRU

$$h_i^1 = \text{GRU}^1(\mathbf{x_i}, h_{i-1}^1), \tag{2}$$

where the function $GRU^1$ is defined as,

$$
\begin{aligned}
z_i &= \sigma(W_z[\mathbf{x_i}, h_{i-1}^1]) \\
r_i &= \sigma(W_r[\mathbf{x_i}, h_{i-1}^1]) \\
\widetilde{h}_i^1 &= tanh(W_h[\mathbf{x_i}, r_i \odot h_{i-1}^1]) \\
h_i^1 &= (1 - z_i) \odot h_{i-1}^1 + z_i \odot \widetilde{h}_i^1
\end{aligned}
\tag{3}
$$

It consists of two gatings $z_i, r_i$, controlling whether the current hidden state $h_i^1$ should be directly copied from $h_{i-1}^1$ or should pass through a more complex path $\widetilde{h}_i^1$.

Given the sentence feature vector $h_n^1$, we then compute an importance weight vector $\alpha_i$ of each word for the second reading. We put the importance weight $\alpha_i$ on the skip-connections as shown in Fig. 2(a) to bias the two information flows: If the current word $x_i$ has a very small weight $\alpha_i$, then the second read hidden state $h_i^2$ will mostly take the information directly from the previous state $h_{i-1}^2$, ignoring the influence of the current word. If $\alpha_i$ is close to 1 then it will be similar to a standard GRU, which is only influenced from the current word. Thus the second reading has the following update rule

$$h_i^2 = (1 - \alpha_i) \odot h_{i-1}^2 + \alpha_i \odot \text{GRU}^2(\mathbf{x_i}, h_{i-1}^2), \tag{4}$$

where $\odot$ means element-wise product. We compute the importance weights by attending $h_i^1$ with $h_n^1$ as follows

$$\alpha_i = tanh(W_e h_i^1 + U_e h_n^1 + V_e \mathbf{x_i}), \tag{5}$$

where $W_e$, $U_e$, $V_e$ are learnable parameters. Note that $\alpha_i$ is a vector representing the importance of each dimension in the word embedding. Empirically, we find that using a vector is better than a scalar gating. We hypothesize that this is because different dimensions represent different semantic meanings, and a scalar gating mechanism lacks the ability to capture the variances among these dimensions.

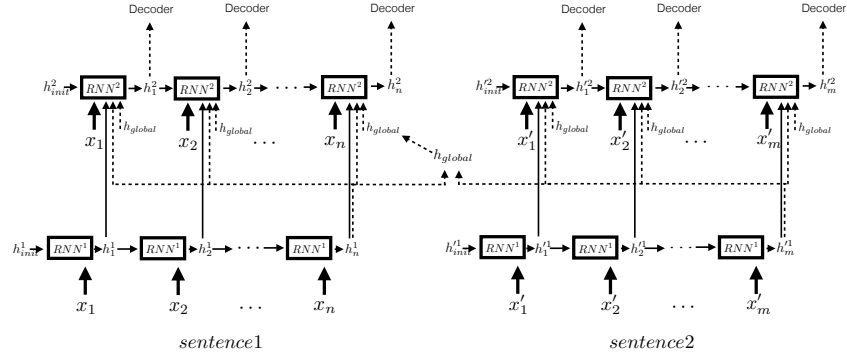

Figure 3: Hierachical Read-Again

Combining this with the standard GRU update rule

$$\text{GRU}^2(\mathbf{x_i}, h_{i-1}^2) = (1 - z_i) \odot h_{i-1}^2 + z_i \odot \widetilde{h}_i^2,$$

we can simplify the updating rule Eq. (4) to get

$$h_i^2 = (1 - \alpha_i \odot z_i) \odot h_{i-1}^2 + (\alpha_i \odot z_i) \odot \widetilde{h}_i^2 \qquad (6)$$

This equations shows that our 'read-again' model on GRU is equivalent to replace the GRU cell with a more general gating mechanism that also depends on the feature representation of the whole sentence computed from the first reading pass. We argue that adding this global information could help direct the information flow for the forward pass resulting in a better encoder.

### 3.1.2 LSTM READ-AGAIN

We now apply the 'Read-Again' idea to the LSTM architecture as shown in Fig. 2(b). Our first reading is performed by an $LSTM^1$ defined as

$$f_i = \sigma(W_f[\mathbf{x_i}, h_{i-1}]) \qquad (7)$$
$$i_i = \sigma(W_i[\mathbf{x_i}, h_{i-1}])$$
$$o_i = \sigma(W_o[\mathbf{x_i}, h_{i-1}])$$
$$\widetilde{C_i} = tanh(W_C[\mathbf{x_i}, h_{i-1}])$$
$$C_i = f_t \odot C_{i-1} + i_i \odot \widetilde{C_i}$$
$$h_i = o_i \odot tanh(C_i)$$

Different from the GRU architecture, LSTM calculates the hidden state by applying a non-linear activation function to the cell state $C_i$, instead of a linear combination of two paths used in the GRU. Thus for our second read, instead of using skip-connections, we make the gating functions explicitly depend on the whole sentence vector computed from the first reading pass. We argue that this helps the encoding of the second reading $LSTM^2$, as all gating and updating increments are also conditioned on the whole sequence feature vector $(h_i^1, h_n^1)$. Thus

$$h_i^2 = \text{LSTM}^2([\mathbf{x_i}, h_i^1, h_n^1], h_{i-1}^2), \qquad (8)$$

### 3.1.3 READING MULTIPLE SENTENCES

In this section we extend our 'Read-Again' model to the case where the input sequence has more than one sentence. Towards this goal, we propose to use a hierarchical representation, where each sentence has its own feature vector from the first reading pass. We then combine them into a single vector to bias the second reading pass. We illustrate this in the context of two input sentences, but it is easy to generalize to more sentences. Let $\{x_1, x_2, \cdots, x_n\}$ and $\{x'_1, \cdots, x'_m\}$ be the two

input sentences. The first RNN reads these two sentences independently to get two sentence feature vectors $h_n^1$ and $h_m'^1$ respectively.

Here we investigate two different ways to handle multiple sentences. Our first option is to simply concatenate the two feature vectors to bias our second reading pass:

$$h_i^2 = \text{RNN}^2([\mathbf{x_i}, h_i^1, h_n^1, h_m'^1], h_{i-1}^2) \tag{9}$$

$$h_i'^2 = \text{RNN}^2([\mathbf{x_i'}, h_i'^1, h_n^1, h_m'^1], h_{i-1}'^2)$$

where $h_0^2$ and $h_0'^2$ are initialized as zero vectors. Feeding $h_n^1, h_m'^1$ into the second RNN provides more global information explicitly and helps acquire long term dependencies.

The second option we explored is shown in Fig. 3. In particular, we use a non-linear transformation to get a single feature vector $h_{global}$ from both sentence feature vectors:

$$h_{global} = tanh(W_r h_n^1 + U_r h_m'^1 + v_r) \tag{10}$$

The second reading pass is then

$$\widetilde{h_i^2} = \text{RNN}^2([\mathbf{x_i}, h_i^1, h_n^1, h_{global}], h_{i-1}^2) \tag{11}$$

$$\widetilde{h_i'^2} = \text{RNN}^2([\mathbf{x_i'}, h_i'^1, h_m'^1, h_{global}], h_{i-1}'^2)$$

Note that this is more easily scalable to more sentences. In our experiments both approaches perform similarly.

## 3.2 DECODER WITH COPY MECHANISM

In this paper we argue that only a small number of common words are needed for generating a summary in addition to the words that are present in the source text. We can consider this as a hybrid approach which combines extractive and abstractive summarization. This has two benefits: first it allow us to use a very small vocabulary size, speeding up inference. Furthermore, we can create summaries which contain OOVs if they are present in the source text.

Our decoder reads the vector representations of the input text using an attention mechanism, and generates the target summary word by word. We use an LSTM as our decoder, with a fixed-size vocabulary dictionary $Y$ and learnable word embeddings $\mathbf{Y} \in \mathbf{R}^{|Y| \times dim}$. At time-step $t$ the LSTM generates a summary word $y_t$ by first computing the current hidden state $s_t$ from the previous hidden state $s_{t-1}$, previous summary word $y_{t-1}$ and current context vector $c_t$

$$s_t = LSTM([\mathbf{y_{t-1}}, c_t], s_{t-1}), \tag{12}$$

where the context vector $c_t$ is computed with an attention mechanism on the encoder hidden states:

$$c_t = \sum_{i=1}^{n} \beta_{it} h_i^2. \tag{13}$$

The attention score $\beta_{it}$ at time-step $t$ on the $i$-th word is computed via a soft-max over $o_{it}$, where

$$o_{it} = att(s_{t-1}, h_i^2) = v_a^T tanh(W_a s_{t-1} + U_a h_i^2), \tag{14}$$

with $v_a, W_a, U_a$ learnable parameters.

A typical way to treat OOVs is to encode them with a single shared embedding. However, different OOVs can have very different meanings, and thus using a single embedding for all OOVs will confuse the model. This is particularly detrimental when using small vocabulary sizes. Here we address this issue by deriving the representations of OOVs from their corresponding context in the input text. Towards this goal, we change the update rule of $\mathbf{y_{t-1}}$. In particular, if $y_{t-1}$ belongs to a word that is in our decoder vocabulary we take its representation from the word embedding, otherwise if it appears in the input sentence as $x_i$ we use

$$\mathbf{y_{t-1}} = \mathbf{p_i} = tanh(W_c h_i^2 + b_c) \tag{15}$$

where $W_c$ and $b_c$ are learnable parameters. Since $h_i^2$ encodes useful context information of the source word $x_i$, $p_i$ can be interpreted as the semantics of this word extracted from the input sentence. Furthermore, if $y_{t-1}$ does not appear in the input text, nor in $Y$, then we represent $\mathbf{y_{t-1}}$ using the <UNK> embedding.

Given the current decoder's hidden state $s_t$, we can generate the target summary word $y_t$. As shown in Fig. 1(b), at each time step during decoding, the decoder outputs a distribution over generating words from $Y$, as well as over copying a specific word $x_i$ from the source sentence.

### 3.3 LEARNING

We jointly learn our encoder and decoder by maximizing the likelihood of decoding the correct word at each time step. We refer the reader to the experimental evaluation for more details.

## 4 EXPERIMENTAL EVALALUATION

In this section, we show results of abstractive summarization on Gigaword (Graff & Cieri (2003); Napoles et al. (2012)) and DUC2004 (Over et al. (2007)) datasets. Our model can learn a meaningful re-reading weight distribution for each word in the input text, putting more emphasis on important verb and nous, while ignoring common words such as prepositions. As for the decoder, we demonstrate that our copy mechanism can successfully reduce the typical vocabulary size by a factor 5 while achieving much better performance than the state-of-the-art, and by a factor of 30 while maintaining the same level of performance. In addition, we provide an analysis and examples of which words are copied during decoding.

**Dataset and Evaluation Metric:** We use the Gigaword corpus to train and evaluate our models. Gigaword is a news corpus where the title is employed as a proxy for the summary of the article. We follow the same pre-processing steps of Rush et al. (2015), which include filtering, PTB tokenization, lower-casing, replacing digit characters with #, replacing low-frequency words with UNK and extracting the first sentence in each article. This results in a training set of 3.8M articles, a validation set and a test set each containing 400K articles. The average sentence length is 31.3 words for the source, and 8.3 words for the summaries. Following the standard protocol we evaluate ROUGE score on 2000 random samples from the test set. As for evaluation metric, we use full-length F1 score on Rouge-1, Rouge-2 and Rouge-L, following Chopra et al. (2016) and Nallapati et al. (2016), since these metrics are less bias to the outputs' length than full-length recall scores.

**Implemetation Details:** We implement our model in Tensorflow and conduct all experiments on a NVIDIA Titan X GPU. Our models converged after 2-3 days of training, depending on model size. Our RNN cells in all models have 1 layer, 512-dimensional hidden states, and 512-dimensional word embeddings. We use dropout rate of 0.2 in all activation layers. All parameters, except the biases are initialized uniformly with a range of $\sqrt{3/d}$, where $d$ is the dimension of the hidden state (Sussillo & Abbott (2014)). The biases are initialized to 0.1. We use plain SGD to train the model with gradient clipped at 10. We start with an initial learning rate of 2, and halve it every epoch after first 5 epochs. Our max epoch for training is 10. We use a mini-batch size of 64, which is shuffled during training.

### 4.1 QUANTITATIVE EVALUATION

| #Input | Model | Size | Rouge-1 | Rouge-2 | Rouge-L |
|--------|-------|------|---------|---------|---------|
| 1 sent | ABS (baseline) | 69K | 24.12 | 10.24 | 22.61 |
|  | GRU (baseline) | 69K | 26.79 | 12.03 | 25.14 |
|  | Ours-GRU | 69K | 27.26 | 12.28 | 25.48 |
|  | Ours-LSTM | 69K | **27.82** | **12.74** | **26.01** |
|  | GRU (baseline) | 15K | 24.67 | 11.30 | 23.28 |
|  | Ours-GRU | 15K | 25.04 | 11.40 | 23.47 |
|  | Ours-LSTM | 15K | 25.30 | 11.76 | 23.71 |
|  | Ours-GRU (C) | 15K | **27.41** | 12.58 | **25.74** |
|  | Ours-LSTM (C) | 15K | 27.37 | **12.64** | 25.69 |
| 2 sent | Ours-Opt-1 (C) | 15K | 27.95 | **12.65** | 26.10 |
|  | Ours-Opt-2 (C) | 15K | **27.96** | 12.65 | **26.18** |

Table 1: Different Read-Again Model. Ours denotes Read-Again models. C denotes copy mechanism. Ours-Opt-1 and Ours-Opt-2 are the models described in section 3.1.3. Size denotes the size of decoder vocabulary in a model.

**Results on Gigaword:** We compare the performances of different architectures and report ROUGE scores in Table 1. Our baselines include the ABS model of Rush et al. (2015) with its proposed

vocabulary size as well as an attention encoder-decoder model with uni-directional GRU encoder. We allow the decoder to generate variable length summaries. As shown in Table 1 our Read-Again models outperform the baselines on all ROUGE scores, when using both 15K and 69K sized vocabularies. We also observe that adding the copy mechanism further helps to improve performance: Even though the decoder vocabulary size of our approach with copy (15K) is much smaller than ABS (69K) and GRU (69K), it achieves a higher ROUGE score. Besides, our Multiple-Sentences model achieves the best performance.

| Models | Size | Rouge-1 | Rouge-2 | Rouge-L |
|---|---|---|---|---|
| ZOPIARY (Zajic et al. (2004)) | - | 25.12 | 6.46 | 20.12 |
| ABS (Rush et al. (2015)) | 69K | 26.55 | 7.06 | 23.49 |
| ABS+ (Rush et al. (2015)) | 69K | 28.18 | 8.49 | 23.81 |
| RAS-LSTM (Chopra et al. (2016)) | 69K | 27.41 | 7.69 | 23.06 |
| RAS-Elman (Chopra et al. (2016)) | 69K | 28.97 | 8.26 | 24.06 |
| big-words-lvt2k-1sent (Nallapati et al. (2016)) | 69K | 28.35 | **9.46** | 24.59 |
| big-words-lvt5k-1sent (Nallapati et al. (2016)) | 200K | 28.61 | 9.42 | 25.24 |
| Ours-GRU (C) | 15K | 29.08 | 9.20 | 25.25 |
| Ours-LSTM (C) | 15K | **29.89** | 9.37 | 25.93 |
| Ours-Opt-2 (C) | 15K | 29.74 | 9.44 | **25.94** |

Table 2: Rouge-N limited-length recall on DUC2004. Size denotes the size of decoder vocabulary in a model.

**Evaluation on DUC2004:** DUC 2004 (Over et al. (2007)) is a commonly used benchmark on summarization task consisting of 500 news articles. Each article is paired with 4 different human-generated reference summaries, capped at 75 characters. This dataset is evaluation-only. Similar to Rush et al. (2015), we train our neural model on the Gigaword training set, and show the models' performances on DUC2004. Following the convention, we also use ROUGE limited-length recall as our evaluation metric, and set the capping length to 75 characters. We generate summaries with 15 words using beam-size of 10. As shown in Table 2, our method outperforms all previous methods on Rouge-1 and Rouge-L, and is comparable on Rouge-2. Furthermore, our model only uses 15k decoder vocabulary, while previous methods use 69k or 200k.

**Importance Weight Visualization:** As we described in the section before, $\alpha_i$ is a high-dimension vector representing the importance of each word $x_i$. While the importance of a word is different over each dimension, by averaging we can still look at general trends of which word is more relevant. Fig. 4 depicts sample sentences with the importance weight $\alpha_i$ over input words. Words such as *the*, *a*, *'s*, have small $\alpha_i$, while words such as *aeronautics, resettled, impediments,* which carry more information have higher values. This shows that our read-again technique indeed extracts useful information from the first reading to help bias the second reading results.

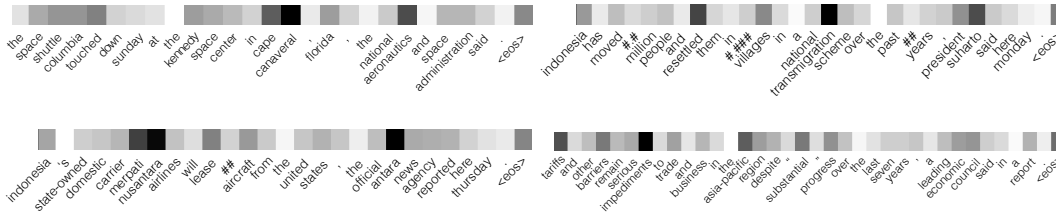

Figure 4: Weight Visualization. Black indicates high weight

## 4.2 EVALUATION OF COPY MECHANISM

Table 3 shows the effect on our model of decreasing the decoder vocabulary size. We can see that when using the copy mechanism, we are able to reduce the decoder vocabulary size from 69K to 2K, with only 2-3 points drop on ROUGE score. This contrasts the models that do not use the copy mechanism. Equipped with a copy mechanism, our model is able to generate OOVs as summary

| Size | Rouge-1 | | Rouge-2 | | Rouge-L | |
|---|---|---|---|---|---|---|
| | Ours-LSTM | Ours-LSTM (C) | Ours-LSTM | Ours-LSTM (C) | Ours-LSTM | Ours-LSTM (C) |
| 2K | 14.39 | 24.21 | 6.46 | 11.27 | 13.74 | 23.09 |
| 5K | 20.61 | 26.83 | 9.67 | 12.66 | 19.58 | 25.31 |
| 15K | 25.30 | 27.37 | 11.76 | 12.64 | 23.74 | 25.69 |
| 30K | 26.86 | 27.49 | 11.93 | 12.75 | 25.16 | 25.77 |
| 69K | 27.82 | 27.89 | 12.73 | 12.69 | 26.01 | 26.03 |

Table 3: ROUGE Evaluation for Models with Different Decoder Size and 110k Encoder Size. Ours denotes Read-Again. C denotes copy mechanism.

| Size | Rouge-1 | | Rouge-2 | | Rouge-L | |
|---|---|---|---|---|---|---|
| | Ours-LSTM | Ours-LSTM (C) | Ours-LSTM | Ours-LSTM (C) | Ours-LSTM | Ours-LSTM (C) |
| 5K | 21.82 | 26.57 | 9.80 | 11.98 | 20.60 | 25.00 |
| 15K | 23.84 | 27.79 | 10.69 | 12.54 | 22.50 | 25.96 |
| 30K | 23.78 | 27.48 | 10.68 | 12.56 | 22.28 | 25.94 |
| 110K | 25.30 | 27.37 | 11.76 | 12.64 | 23.74 | 25.69 |

Table 4: ROUGE Evaluation for Models with Different Encoder Size and 15k Decoder Size. Ours denotes Read-Again. C denotes copy mechanism.

words, and thus maintains its expressive ability even with a small decoder vocabulary size. We also observe from Table 4 that the copy mechanism help us to decrease the encoder vocabulary size as well. The model without copy suffers from severe OOV problem when encoder size is small, since a single shared <UNK> embedding cannot depict many different OOVs. This makes it difficult for the encoder to understand the input text. Meanwhile, our copy model can extract an OOV's meaning accordingly from its context in the input text, and thus it is sufficient to learn and store only the high-frequency words embeddings using our model, which in turn save the storage. We also notice that shrinking the encoder vocabulary to 15k achieves better result. One possible reason is that long tail words can not learn efficient embeddings during training, and representing them with extracted embedding from our model performs better.

Table 5 shows the decoding time as a function of vocabulary size. As computing the soft-max is usually the bottleneck for decoding, reducing vocabulary size dramatically reduces the decoding time from 0.38 second per sentence to 0.08 second.

| Decoder-Size | 2k | 5k | 15k | 30k | 69k |
|---|---|---|---|---|---|
| Ours-LSTM | 0.076 | 0.081 | 0.111 | 0.161 | 0.356 |
| Ours-LSTM (C) | 0.084 | 0.090 | 0.123 | 0.171 | 0.376 |

Table 5: Decoding Time (s) per Sentence of Models with Different Decoder Size

Table 6 provides some examples of visualization of the copy mechanism. Note that we are able to copy key words from source sentences to improve the summary. From these examples we can see that our model is able to copy different types of rare words, such as special entities' names in case 1 and 2, rare nouns in case 3 and 4, adjectives in case 5 and 6, and even rare verbs in the last example. Note that in the third example, when the copy model's decoder uses the embedding of *headmaster* as its first input, which is extracted from the source sentence, it generates the same following sentence as the no-copy model. This probably means that the extracted embedding of *headmaster* is closely related to the learned embedding of *teacher*.

| Input: | air new zealand said friday it had reached agreement to buy a ## percent interest in australia 's ***ansett*** holdings limited for ### million australian -lrb- ### million us dollars -rrb- . |
|---|---|
| **Golden:** | urgent air new zealand buys ## percent of australia 's ansett airlines |
| **No Copy:** | air nz to buy ## percent stake in australia 's <unk> |
| **Copy:** | air nz to buy ## percent stake in ***ansett*** |
| **Input:** | yemen 's ruling party was expected wednesday to nominate president ali abdullah saleh as its candidate for september 's presidential election , although ***saleh*** insisted he is not bluffing about bowing out. |
| **Golden:** | the #### gmt news advisory |
| **No Copy:** | yemen 's ruling party expected to nominate president as presidential candidate |
| **Copy:** | yemen 's ruling party expected to nominate ***saleh*** as presidential candidate |
| **Input:** | a ##-year-old ***headmaster*** who taught children in care homes for more than ## years was jailed for ## years on friday after being convicted of ## sexual assaults against his pupils. |
| **Golden:** | britain : headmaster jailed for ## years for paedophilia |
| **No Copy:** | teacher jailed for ## years for sexually abusing childre |
| **Copy:** | ***headmaster*** jailed for ## years for sexually abusing children |
| **Input:** | singapore 's rapidly ***ageing*** population poses the major challenge to fiscal policy in the ##st century , finance minister richard hu said , and warned against european-style state <unk>. |
| **Golden:** | ageing population to pose major fiscal challenge to singapore |
| **No Copy:** | finance minister warns against <unk> state |
| **Copy:** | s pore 's ***ageing*** population poses challenge to fiscal policy |
| **Input:** | angola is planning to ***refit*** its ageing soviet-era fleet of military jets in russian factories , a media report said on tuesday. |
| **Golden:** | angola to refit jet fighters in russia : report |
| **No Copy:** | angola to <unk> soviet-era soviet-era fleet |
| **Copy:** | angola to ***refit*** military fleet in russia |

Table 6: Visualization of Copy Mechanism

## 5 CONCLUSION

In this paper we have proposed two simple mechanisms to alleviate the problems of current encoder-decoder models. Our first contribution is a 'Read-Again' model which does not form a representation of the input word until the whole sentence is read. Our second contribution is a copy mechanism that can handle out-of-vocabulary words in a principled manner allowing us to reduce the decoder vocabulary size and significantly speed up inference. We have demonstrated the effectiveness of our approach in the context of summarization and shown state-of-the-art performance. In the future, we plan to tackle summarization problems with large input text. We also plan to exploit our findings in other tasks such as machine translation.

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
