# Peer review of "Efficient Summarization with Read-Again and Copy Mechanism"

_ICLR 2017 — rejected_

[Public Comment · Ehud Ben-Reuven · 12 Nov 2016]
**can you please comment on how p_t is used in the copy mechanisem**

Please see

[Official Review · AnonReviewer2 · rating 5 · confidence 5 · 16 Dec 2016]
**Official Review**

This work explores the neural models for sentence summarisation by using a read-again attention model and a copy mechanism which grants the ability of direct copying word representations from the source sentences. The experiments demonstrate the model achieved better results on DUC dataset. Overall, this paper is not well-written. There are confusing points, some of the claims are lack of evidence and the experimental results are incomplete. 

Detailed comments:
 
-Read-again attention. How does it work better than a vanilla attention? What would happen if you read the same sentences multiple times? Have you compared it with staked LSTM (with same number of parameters)? There is no model ablation in the experiment section. 

-Why do you need reading two sentences? The Gigaword dataset is a source-to-compression dataset which does not need multiple input sentences. How do you compare your model with single sent input and two sent input?

-Copy mechanism. What if there are multiple same words appeared in the source sentences to be copied? According to equation (5), you only copy one vector to the decoder. However, there is no this kind of issue for a hard copy mechanism. Besides, there is no comparison between the hard copy mechanism and this vector copy mechanism in the experiment section

-Vocabulary size. This part is a bit off the main track of this paper. If there is no evidence showing this is the special property of vector copy mechanism, it would be trivial in this paper. 

-Experiments. On the DUC dataset, it compares the model with other up-to-date models, while on the Gigaword dataset paper only compares the model with the ABS Rush et al. (2015) and the GRU (?), which are quite weak baseline models. It is irresponsible to claim this model achieved the state-of-the-art performance in the context of summarization.

Typos: (1) Tab. 1. -> Table 1. (2) Fig. 3.1.2.?

[Official Review · AnonReviewer1 · rating 5 · confidence 4 · 19 Dec 2016]
**Review for the Efficient Summarization with Read-Again ...**

Summary: This paper proposes a read-again attention-based representation of the document with the copy mechanism for the summarization task. The model reads each sentence in the input document twice and creates a hierarchical representation of it instead of a bidirectional RNN. During the decoding, it uses the representation of the document obtained via the read-again mechanism and points the words that are OOV in the source document. The model does abstractive summarization. The authors show improvements on DUC 2004 dataset and provide an analysis of their model with different configurations.

Contributions:
The main contribution of this paper is the read-again attention mechanism where the model reads the same sentence twice and obtains a better representation of the document.

Writing:
The text of this paper needs more work. There are several typos and the explanations of the model/architecture are not really clear, some parts of the paper feel somewhat bloated. 

Pros:
- The proposed model is a simple extension to the model to the model proposed in [2] for summarization.
- The results are better than the baselines.

Cons:
- The improvements are not that large.
- Justifications are not strong enough.
- The paper needs a better writeup. Several parts of the text are not using a clear/precise language and the paper needs a better reorganization. Some parts of the text is somewhat informal.
- The paper is very application oriented.

Question:
- How does the training speed when compared to the regular LSTM?

Some Criticisms:

A similar approach to the read again mechanism which is proposed in this paper has already been explored in [1] in the context of algorithmic learning and I wouldn’t consider the application of that on the summarization task a significant contribution.  The justification behind the read-again mechanism proposed in this paper is very weak. It is not really clear why additional gating alpha_i is needed for the read again stage.
As authors also suggest, pointer mechanism for the unknown/rare words [2] and it is adopted for the read-again attention mechanism. However, in the paper, it is not clear where the real is the gain coming from, whether from “read-again” mechanism or the use of “pointing”. 
The paper is very application focused, the contributions of the paper in terms of ML point of view is very weak.
It is possible to try this read-again mechanism on more tasks other than summarization, such as NMT, in order to see whether if those improvements are 
The writing of this paper needs more work. In general, it is not very well-written. 

Minor comments:

Some of the corrections that I would recommend fixing,

On page 4: “… better than a single value … ” —> “… scalar gating …”
On page 4: “… single value lacks the ability to model the variances among these dimensions.” —> “… scalar gating couldn’t capture the ….”
On page 6: “ … where h_0^2 and h_0^'2 are initial zero vectors … “ —> “… h_0^2 and h_0^'2 are initialized to a zero vector in the beginning of each sequence …"

There are some inconsistencies for example parts of the paper refer to Tab. 1 and some parts of the paper refer to Table 2.

Better naming of the models in Table 1 is needed.
The location of Table 1 is a bit off.

[1] Zaremba, Wojciech, and Ilya Sutskever. "Reinforcement learning neural Turing machines." arXiv preprint arXiv:1505.00521 362 (2015). 
[2] Gulcehre, Caglar, et al. "Pointing the Unknown Words." arXiv preprint arXiv:1603.08148 (2016).

[Official Review · AnonReviewer3 · rating 6 · confidence 4 · 19 Dec 2016]
**Solid incremental work but need better writing and more experiments to be more convincing**

This paper proposed two incremental ideas to extend the current state-of-the-art summarization work based on seq2seq models with attention and copy/pointer mechanisms.

1. This paper introduces 2-pass reading where the representations from the 1st-pass is used to  re-wight the contribution of each word to the sequential representation of the 2nd-pass. The authors described how such a so-called read-again process applies to both GRU and LSTM.
 
2. On the decoder side, the authors also use the softmax to choose between generating from decoder vocabulary and copying a source position, with a new twist of representing the previous decoded word Y_{t-1} differently. This allows the author to explore a smaller decoder vocabulary hence led to faster inference time without losing summarization performance.
 
This paper claims the new state-of-the-art on DUC2004 but the comparison on Gigaword seems to be incomplete (missing more recent results after Rush 2015 etc). While the overall work is solid, there are also other things missing scientifically. For example, 
- how much computational costs does the 2nd pass reading add to the end-to-end system? 
- How does the decoder small vocabulary trick work without 2nd-pass reading on the encoder side for both summarization performance and runtime speed?
- There are other ways to improve the embedding of a sentence. How does the 2nd-pass reading compare to recent work from multiple authors on self-attention and/or LSTMN? For example, Cheng et al. 2016, Long Short-Term Memory-Networks for Machine Reading; Parikh et al. 2016, A Decomposable Attention Model for Natural Language Inference?

[Public Comment · ICLR 2017 conference · 07 Jan 2017]
**Authors: Please post rebuttal**

Reviewers are currently in discussion. Please post a rebuttal to any comments or questions in their reviews.

Thanks!

[Author Response · Wenyuan Zeng · 16 Jan 2017]
**Extra Experiment for Copy Mechanism**

In the paper, we've shown that our copy mechanism can reduce the decoder size. Here, we show that it can also help to reduce the encoder vocabulary size. We fixed the decoder size as 15K, and we can see that with copy mechanism we are able to reduce encoder vocabulary size from 110k to 15k without loss of performance, while model without copy has severe decreasing. The reason is that our copy mechanism is able to extract an rare word’s embedding from its context, and thus it is enough to learn and store the high-frequency words’ embeddings in our model.

	    Rouge1		Rouge2		    RougeL
size	    nocopy    copy	nocopy	copy    nocopy	    copy
5k	    21.8	    26.2	9.8		12.0	    21.6	    25.0
15k	    23.8	    27.8	10.7		12.5	    22.5	    26.0
30k	    23.8	    27.5	10.7		12.6	    22.3	    26.0
110k    25.3         27.4     11.8	        12.6	    23.7	    25.7

[Final Decision · Program Chairs · 06 Feb 2017]
**ICLR committee final decision**

This work presents a method for reducing the target vocabulary for abstractive summarization by employing a read-again copy mechanism. Reviewers felt that the paper was lacking in several regards particularly focusing on issues clarity and originality. 
 
 Issues raised:
 - Several reviewers focused on clarity/writing issues of the work, highlighting inconsistencies of notation, justifications, and extraneous material. They recommend a rewrite of the work. 
 - The question of originality and novelty is important. The copy mechanism has now been invented many times. Reviewers argue that simply applying this to summary is not enough, or at least not the focus of ICLR. The same question is posed about self-attention.
 - In terms of quality, there are concerns about comparisons to more recent baselines and getting the experiments to run correctly on Gigaword.